# Lightweight Pest Object Detection Model for Complex Economic Forest Tree Scenarios

**DOI:** 10.3390/insects16090959

**Published:** 2025-09-12

**Authors:** Xiaohui Cheng, Xukun Wang, Yanping Kang, Yun Deng, Qiu Lu, Jian Tang, Yuanyuan Shi, Junyu Zhao

**Affiliations:** 1College of Computer Science and Engineering, Guilin University of Technology, Guilin 541004, China; cxiaohui@glut.edu.cn (X.C.); 1020231181@glut.edu.cn (X.W.); 2002078@glut.edu.cn (Y.D.); 2002021@glut.edu.cn (Q.L.); 2Guangxi Key Laboratory of Embedded Technology and Intelligent System, Guilin 541004, China; 3Guangxi Forestry Research Institute, Nanning 530002, China; ttljmy@163.com (J.T.); syyfly@163.com (Y.S.); zjyuyu@126.com (J.Z.)

**Keywords:** economic forest pest control, RT-DETR, feature aggregation, small object detection

## Abstract

Pests in economic forests are often difficult to detect due to their resemblance to backgrounds, and traditional manual monitoring is inefficient and prone to missed detections. This study proposes LightFAD-DETR, a lightweight model specifically designed for pest identification in complex forest scenarios. The model accurately identifies and pinpoints tiny pests while distinguishing their growth stages. Its innovative design significantly reduces computational demands, enabling direct deployment on edge devices. This technology helps reduce excessive pesticide use, maintain tree health, and safeguard yields of fruits and timber, providing practical support for sustainable forestry management.

## 1. Introduction

Economic forest trees, including high-value species such as rubber, tea oil camellia, walnut, chestnut, eucalyptus, poplar, and fruit trees, have become vital pillars for national ecological security barriers, strategic timber resource reserves, and rural revitalization. Their economic value is constituted by providing core income sources for forest farmers and supporting extensive industrial chains encompassing timber processing, forest chemical industries, fruit, and medicinal material production [1,2]. Various pests, such as longhorn beetles, pine caterpillars, scale insects, as well as wood-boring and foliage-feeding insects, pose pervasive and severe threats to these economically important trees, which are characterized by long growth cycles and exposure to open environments. The explosive, sudden, and devastating nature of pest infestations leads to stunted tree growth, deterioration of timber quality, significant yield reduction or even complete loss of fruit, and in severe cases, the death of entire stands. These consequences result in incalculable direct and indirect economic losses [3,4]. Furthermore, traditional control methods reliant on chemical pesticides are not only costly but also incur serious negative impacts, including environmental pollution, pesticide residues, and the destruction of natural enemies [5]. Economic forests often grow in heterogeneous environments that can be described as complex economic forest ecosystems. This concept refers to multi-species stands with diverse canopy structures and variable microclimatic conditions, which not only harbor diverse pest species but also generate intricate visual backgrounds that increase the difficulty of pest monitoring.

Current monitoring methods relying on manual inspections and physicochemical trapping exhibit significant limitations. Manual inspections are highly inefficient, making it difficult to cover vast and complex forest areas. Identification accuracy heavily depends on the inspector’s experience and is highly subjective, easily leading to missed detections and misjudgments. This approach struggles to capture early-stage, localized pest outbreaks, delaying optimal control timing and posing safety hazards in rugged terrain. Physicochemical trapping [6] has a narrow target range, primarily attracting specific pest species and providing limited information. It cannot precisely locate infested plants or quantitatively assess infestation severity, and it requires ongoing maintenance investment. These methods fall short in timeliness, accuracy, coverage, and cost-effectiveness, failing to meet the demands of modern precision forestry. Within intelligent pest monitoring systems, object detection technology [7] plays a pivotal core role. Its fundamental value lies in simultaneously achieving pest species identification and precise location pinpointing, providing an unprecedented richness in information dimensions. This enables the development of automated, intelligent monitoring systems based on ground-based sensing devices and artificial intelligence. Such systems can efficiently process massive volumes of image data, significantly enhance monitoring efficiency and coverage, reduce human error, improve identification accuracy and objectivity, and hold promise for providing early warnings of minor initial damage [8].

Although object detection technology holds great promise, its research and application in the specific field of economic forest pest control still faces significant challenges [9]. The complex and variable backgrounds of forest areas, such as interlaced branches and leaves, uneven lighting, shadows, and different growth stages, can severely interfere with target recognition [10]. In addition, the targets themselves present complex characteristics: pests are often small, morphologically diverse, camouflaged, or mimetic, with damage symptoms that vary widely in form and scale and can easily be confused with diseases or physiological stress [11]. These factors highlight the urgent need to improve the robustness and generalization ability of existing models in complex real-world scenarios. At the same time, deployment on mobile devices demands lightweight models and fast real-time inference. Here, a “lightweight model” refers to a neural network with reduced parameters and computational requirements, enabling real-time inference on resource-constrained devices.

However, most existing detection models still fall short of these requirements. Many lightweight pest detection methods are developed for crop fields or single-species environments, making them difficult to generalize to complex economic forest scenarios. Conventional approaches also struggle to preserve fine-grained details of tiny pests under strong background interference, resulting in high miss rates. Furthermore, although some models achieve competitive accuracy, they typically incur high computational costs, which hinder practical deployment on resource-constrained edge devices.

To overcome these gaps, this study proposes a lightweight detection paradigm tailored for economic forest pests. The planned scientific novelty lies in combining lightweight architectural design with enhanced cross-regional feature modeling and adaptive small-object representation, while maintaining real-time inference efficiency suitable for edge deployment. This integrated design directly addresses the dual challenges of accuracy in complex backgrounds and practicality for field applications, thus providing a new pathway for intelligent forest-pest monitoring. Based on these considerations, we hypothesize that a lightweight detection model integrating re-parameterized multi-branch convolution, adaptive feature aggregation, and efficient normalization strategies can effectively improve the recognition of small pest objects in complex forestry scenes while maintaining real-time inference speed and reduced computational cost.

Based on the RT-DETR framework, this paper proposes a lightweight detection model, LightFAD-DETR, for complex scenarios. Through multi-dimensional collaborative optimization, it achieves an efficient balance between speed, accuracy, and resource consumption. The main contributions of this paper are as follows:A high-efficiency lightweight model, LightFAD-DETR, is proposed for detecting pests in economic forests. This model achieves an excellent balance between detection accuracy and computational efficiency.The RepNCSPELAN4-CAA module is designed using a re-parameterization strategy that maintains a multi-branch convolution structure during training and fuses it into a single-path structure during inference, reducing computational latency. Moreover, the module incorporates one-dimensional strip convolutions to dynamically establish long-range spatial dependencies, enhancing the model’s capability to model cross-regional features of elongated pest targets.A feature aggregation diffusion network is developed, incorporating a dimension-aware selective integration module. This design adaptively integrates deep semantic features with shallow texture details, mitigating information loss for small-scale pest targets and improving the recognition of subtle pest signs under complex leaf texture backgrounds.An improved AIFI module with re-parameterized batch normalization is proposed. This enhancement adopts a progressive fusion approach to integrate dynamic linear components into the normalization process and ensures their fusion with adjacent linear layers during inference, thereby further reducing computational redundancy and optimizing the model’s execution performance on edge computing devices.

The remainder of this paper is organized as follows: Section 2 provides a brief overview of related work. Section 3 presents a detailed description of the LightFAD-DETR model and its internal components. Section 4 offers a comprehensive analysis of the experimental results, including ablation experiments and comparative experiments. Section 5 discusses the performance of the LightFAD-DETR model. Finally, Section 6 concludes the paper.

## 2. Related Work

### 2.1. Pest Object Detection

In this study, the term “pest object” denotes an individual insect or a visible pest-related symptom localized within an image. Early detection and accurate identification of pest infestations are fundamental to the implementation of effective plant protection measures. Object detection technology based on computer vision, with its non-contact and automation potential, is increasingly becoming a central focus in intelligent pest monitoring research. Tian et al. [12] proposed the MD-YOLO model for detecting small Lepidoptera pests on sticky traps. They developed a dual-path feature fusion network that combines feature extraction and feature aggregation paths, along with targeted data augmentation strategies such as random splicing and point scattering. The model is deployed via a cloud server, integrated into an Internet of Things system, and equipped with a cloud service platform based on a B/S architecture, enabling remote pest image detection and early warning.

Dong et al. [13] introduced a lightweight pest detection model called PestLite, which incorporates a Multi-Scale Feature Pooling Module to enhance detection accuracy for agricultural pests while significantly reducing model complexity, making it more suitable for real-time detection scenarios. Ye et al. [14] proposed an improved framework combining YOLOv8 [15] and SAHI [16], which enhances the detection of tiny tea pests through multi-module optimization. The framework achieves high precision while maintaining lightweight and real-time performance.

Yang et al. [17] developed an improved lightweight tea-pest detection model by introducing deformable convolutions to enhance feature extraction, using dynamic attention mechanisms for feature fusion optimization, and designing a novel decoupled head and improved non-maximum suppression algorithm. These improvements significantly enhance the detection accuracy of small and visually similar pest species in complex backgrounds, providing an efficient tool for pest management in tea plantations. Li et al. [18] proposed the IMLL-DETR model for detecting pests and diseases in lychee leaves. They designed the MDGA module to enhance multi-scale feature extraction and contextual fusion, introduced a P2 detection head to retain small object details, and restructured the loss function to improve localization accuracy, thereby significantly boosting performance in complex agricultural environments.

Chen et al. [19] developed the Pest-PVT model, which improves detection accuracy for dense, small-sized pests in the field by integrating anchor-free detection, adaptive sample selection, and split self-attention mechanisms. The study utilized the NVIDIA Jetson TX2 embedded platform, along with the JetPack development kit, CUDA, and the MMdetection framework, and developed a graphical interface using PySimpleGUI to deploy and apply the model on low-power edge devices. Zhao et al. [20] introduced the AC-YOLO model to address the challenge of stored grain pest detection. By integrating attention mechanisms into the YOLOv7 [21] framework, the model improves recognition accuracy for multiple small target categories, providing an effective solution for intelligent pest monitoring in grain storage facilities.

### 2.2. Small Object Detection

As one of the key challenges in the field of computer vision, small object detection has attracted increasing research attention in recent years. A wide range of innovative methods have emerged, with significant progress made in areas such as improving feature pyramids, introducing attention mechanisms, optimizing contextual information, and deploying lightweight models. Despite these advances, the inherent characteristics of small objects such as low resolution, limited information, and high sensitivity to background interference continue to pose considerable challenges. These factors make small object detection a central focus for current research and ongoing optimization. Deng et al. [22] addressed the challenges of feature coupling and information loss in small object detection by proposing an extended feature pyramid network. They added a high-resolution pyramid layer specifically for handling small objects and designed a feature texture transfer module to super-resolve feature maps and extract reliable regional details, effectively improving detection accuracy for small objects.

Jiang et al. [23] tackled core challenges in aerial imagery from drones, where small objects are prevalent, densely packed, and vary greatly in scale. They enhanced small object perception by replacing the standard detection head with a specialized head for tiny objects. A multi-scale feature extraction module was designed using multi-branch heterogeneous convolution to capture rich scale information. A bidirectional dense feature pyramid was also constructed to deeply integrate shallow detail with deep semantic information, greatly enhancing small object detection performance in drone images. Cheng et al. [24] conducted a systematic review of the key challenges in small object detection, including information loss, feature noise, sensitivity to bounding box shifts, and insufficient training samples. They proposed six categories of algorithmic strategies, namely sample-based approaches, scale-aware methods, attention mechanisms, feature mimicry, context modeling, and focused detection strategies. In addition, they built several datasets across related fields to support algorithm evaluation and development.

Wang et al. [25] focused on small object detection in autonomous driving scenarios. They introduced an innovative query-guided module called Q-block, which uses coarse localization to identify key regions and then guides fine-grained detection using high-resolution features. This method significantly improves the localization accuracy of distant small objects. Zhang et al. [26] addressed small object detection in industrial defect inspection. They designed a lightweight feature fusion network sensitive to detail, which improved detection performance for small-scale defects across multiple resolutions while maintaining real-time inference speed. This provides an efficient solution for industrial applications. Shen et al. [27] introduced a dynamic perception module for small objects that treats the feature pyramid as a temporal sequence, similar to video frames. By using three-dimensional convolution, the module focuses on multi-scale features and surrounding context of small objects. This enhances fine detail extraction and improves small object detection performance in remote sensing imagery.

## 3. Proposed Method

As shown in Figure 1, based on the RT-DETR [28] framework, this paper proposes a lightweight detection model called LightFAD-DETR designed for complex scenarios. Through multi-dimensional collaborative optimization, the model achieves an efficient balance between speed, accuracy, and resource consumption. First, the RepNCSPELAN4-CAA module employs re-parameterization technology to merge the multi-branch convolutions used during training into an equivalent single-path structure during inference, significantly reducing computational latency. At the same time, it integrates one-dimensional strip convolutions to dynamically capture long-range spatial dependencies, enhancing the model’s ability to represent cross-region associations for elongated pest targets. Second, a feature aggregation diffusion network (FADN) is designed, incorporating a dimension-aware selective integration (DASI) module. This enables dynamic fusion of deep semantic features and shallow detail features, alleviating information loss for small targets, which is especially beneficial for recognizing subtle pest signs under leaf texture interference. In addition, an improved AIFI module with re-parameterized batch normalization (RepBN) is introduced. Through a progressive strategy, dynamic linear components are integrated into the normalization process and fused with adjacent linear layers during inference, further eliminating redundant computations and enhancing efficiency for edge deployment. These innovative designs significantly reduce the model’s parameter count and computational complexity while maintaining high-resolution image processing capability, making it particularly suitable for resource-constrained edge devices.

As shown in Figure 1, based on the RT-DETR model, we first incorporate the first nine layers of the YOLOv9 [29] backbone to replace the original backbone of RT-DETR, which helps reduce the number of parameters and computational cost. Next, the improved RepNCSPELAN4-CAA module is used to replace the RepNCSPELAN4 module in the backbone and the RepC3 module in the CCFM component of RT-DETR. By applying re-parameterization techniques, the multi-branch structure used during training is merged into a single-path structure during inference, thereby reducing computational overhead. Then, a feature aggregation diffusion network is designed, and the DASI module is introduced to enhance the preservation of small object details and improve resistance to background interference. Finally, the AIFI module is improved with the integration of RepBN, forming the new RepBN-AIFI module, which helps reduce training fluctuations and improves inference speed. These components together form the LightFAD-DETR model. LightFAD-DETR effectively reduces both missed and false detections in complex scenarios, while balancing lightweight design, real-time performance, and accuracy, providing reliable technical support for intelligent forest-pest detection.

### 3.1. RepNCSPELAN4-CAA Module

The RepNCSPELAN4 module is an innovative design in YOLOv9 that deeply integrates re-parameterization techniques with cross-scale feature aggregation. It achieves efficient feature extraction and fusion through a multi-path collaborative mechanism. As shown in Figure 2, the module first expands the input feature channels using a 1 × 1 convolution and splits them into two paths. The main path adopts a re-parameterized cross-stage partial network (RepNCSP), which enhances feature representation through multi-branch convolutions during training and automatically merges them into an equivalent single-path structure during inference. This reduces computational latency while maintaining accuracy. The auxiliary path refines multi-level features through cascaded RepNCSP sub-modules. Each sub-module progressively enhances local features via channel compression, re-parameterized bottleneck layers, and residual connections. Finally, shallow detail features and deep semantic features are densely fused along the channel dimension. The output is then passed through a 1 × 1 convolution to compress the channels, resulting in a multi-scale feature map that combines rich spatial details with strong semantic representation. This module overcomes the representational limitations of traditional convolution by using re-parameterization techniques. It balances computational efficiency and feature diversity through a dynamic channel-splitting strategy and excels at capturing spatial topological relationships of multi-scale objects in complex backgrounds, providing a robust feature foundation for real-time high-precision detection.

The core advantage of the RepNCSPELAN4 module lies in its triple collaborative design of re-parameterization, cross-stage channel optimization, and multi-scale feature aggregation, enabling an efficient balance among speed, accuracy, and lightweight design in pest detection tasks. Compared to traditional modules, it significantly enhances feature representation through multi-branch convolutions during training, which are automatically merged into a single-branch structure during inference. This allows for a notable performance boost with virtually no additional computational cost. Its unique cross-stage channel splitting strategy divides the input features into a main path and a lightweight auxiliary path. The main path deepens feature abstraction through stacked re-parameterized bottleneck layers, while the auxiliary path preserves original information, effectively avoiding gradient redundancy and reducing computational load. This makes it especially suitable for deployment on edge devices. The ELAN-style multi-scale dense feature concatenation mechanism fuses shallow details with deep semantic information, equipping the model with strong discriminative capability for tiny pest objects, densely arranged objects, and complex backgrounds. While maintaining a lightweight structure, this module addresses the limitations of traditional models in real-time scenarios, such as missed or false detections of multi-scale objects, and introduces a novel design paradigm for efficient detection in high-resolution imagery.

Cai et al. [30] proposed the Context Anchor Attention (CAA) module, as shown in Figure 3.

The CAA module performs context modeling through multi-stage collaborative processing. First, the input feature X undergoes global average pooling, denoted as $P_{avg}$, to compress spatial information. Then, a 1 × 1 convolution is applied to adjust the channels, generating an initial global context representation $F^{pool}$, which provides a semantic foundation for subsequent long-range modeling, as shown in Equation (1) [30]:(1)$$F^{pool}={Conv}_{1\times 1}\left(P_{avg}\left(X\right)\right)$$

Subsequently, depthwise separable one-dimensional strip convolutions in the horizontal $\left(1\times k_{b}\right)$ and vertical $\left(k_{b}\times 1\right)$ directions are applied to model spatial dependencies along the width and height dimensions, respectively. The kernel size is dynamically adjusted as $k_{b}=11+2n$, where n represents the network depth, allowing the receptive field to be extended with linear complexity and better adapted to elongated object structures. As shown in Equations (2) and (3) [30]:(2)$$F^{w}={DWConv}_{1\times k_{b}}\left(F^{pool}\right)$$(3)$$F^{h}={DWConv}_{k_{b}\times 1}\left(F^{w}\right)$$

Finally, the direction-sensitive feature $F^{h}$ is mapped into a spatial attention map A. A lightweight convolution activated by the Sigmoid function generates spatial attention weights, which are dynamically fused with the local feature P through weighted integration. A residual connection is then applied to enhance the response in target regions, resulting in the output enhanced feature $F^{attn}$. As shown in Equations (4) and (5) [30]:(4)$$A=Sigmoid\left({Conv}_{1\times 1}\left(F^{h}\right)\right)$$(5)$$F^{attn}=\left(A⨀P\right)⊕P$$

Here, “⨀” denotes element-wise multiplication, and “⨁” denotes element-wise addition.

This process decomposes large-kernel convolutions into lightweight one-dimensional operations, which reduces computational complexity while accurately modeling cross-region associations of elongated objects in pest images. It also suppresses background noise and achieves a balanced representation of local details and global semantics.

The CAA module dynamically captures long-range contextual information in pest images using lightweight one-dimensional strip convolutions. It combines global average pooling to extract global features and leverages an adaptive attention mechanism to enhance feature responses in object regions. Its core advantage lies in replacing traditional large-kernel convolutions with depthwise separable convolutions in the horizontal and vertical directions. This significantly reduces computational cost while effectively addressing the issues of sparse context in large-scale objects and background noise in small objects. It is particularly effective at handling elongated targets such as long-bodied pests, achieving a balance between local detail preservation and global dependency modeling. The module provides efficient feature enhancement for multi-scale object detection in complex scenarios.

We embed the CAA module between the multi-scale feature concatenation and the final convolution in the RepNCSPELAN4 module. This integration uses a dynamic context anchoring strategy to further enhance the semantic awareness of feature fusion, thereby forming the RepNCSPELAN4-CAA module in LightFAD-DETR. This design retains the original advantages of re-parameterization and cross-stage channel optimization while addressing the challenges of extreme object scale variation and complex background interference in economic forest scenarios. The CAA module’s one-dimensional strip convolutions adaptively capture long-range cross-region dependencies, and the spatial attention mechanism softly filters the concatenated multi-level features, focusing on strengthening the response in target regions while suppressing background noise.

### 3.2. Feature Aggregation Diffusion Network Integrated with Dimension-Aware Selective Integration Module

Xu et al. [31] proposed the DASI module, as shown in Figure 4. The DASI module adaptively selects and fuses multi-level features, dynamically balancing high-dimensional semantic information with low-dimensional detail features, thereby enhancing the saliency of small pest objects.

The DASI module first aligns the high-dimensional feature $F_{h}\in R^{H_{h}\times W_{h}\times C_{h}}$, the low-dimensional feature $F_{l}\in R^{H_{l}\times W_{l}\times C_{l}}$, and the current layer feature $F_{u}\in R^{H\times W\times C}$ to the same spatial size. Then, these features are evenly divided into four parts along the channel dimension. As shown in Equation (6) [31]:(6)$$h_{i},\;l_{i},\;u_{i}=split\left(F_{h}\right),\;split\left(F_{l}\right),\;split\left(F_{u}\right),\;i=1,2,3,4$$

Here, each split has the shape $R^{H\times W\times \frac{C}{4}}$, ensuring that features from different levels are aligned in both spatial and channel dimensions.

Subsequently, each split of the current layer feature $u_{i}$ is passed through a Sigmoid function to generate the weight α, which is then used to dynamically fuse high-dimensional contextual features with low-dimensional detail features based on the value of α. As shown in Equation (7) [31]:(7)$$\alpha =sigmoid\left(u_{i}\right)$$

Based on the weight α, the high-dimensional feature $h_{i}$ and the low-dimensional feature $l_{i}$ are linearly fused. As shown in Equation (8) [31]:(8)$$u_{i}^{\prime }=\alpha ⨀l_{i}+\left(1-\alpha \right)⨀h_{i}$$

Here, ⨀ denotes element-wise multiplication. When α>0.5, the model favors low-dimensional details; otherwise, it focuses on high-dimensional context, enabling task-driven adaptive selection.

Finally, the fused features are concatenated and passed through a convolution to integrate channel information and enhance non-linearity. As shown in Equations (9) and (10) [31]:(9)$$F_{u}^{\prime \prime }=\left[u_{1}^{\prime },u_{2}^{\prime },u_{3}^{\prime },u_{4}^{\prime }\right]$$(10)$$F_{u}^{\prime }=ReLU\left(BN\left(Conv\left(F_{u}^{\prime \prime }\right)\right)\right)$$

Here, the convolution compresses the channels and fuses information across the splits, while batch normalization (BN) and ReLU further optimize the feature distribution and enhance the representational capability of the module.

This module, through lightweight channel splitting and an adaptive weighting mechanism, effectively alleviates the problem of small object information loss caused by multiple downsampling operations while reducing the number of parameters. It achieves efficient collaborative modeling of detail and semantics under complex backgrounds.

The data focused by the DASI module is further processed through a specific diffusion mechanism, as shown in the feature aggregation diffusion network part of the model in Figure 1. The DASI module accepts inputs from three different scales to extract multi-scale texture features with varying receptive fields. Through the diffusion mechanism, features enriched with contextual information are propagated across all detection scales. By performing feature aggregation and diffusion twice, each scale’s features are enhanced with detailed contextual information, which is more beneficial for subsequent object detection and classification. This process forms the feature aggregation diffusion network.

To address the challenges posed by pest objects that are small in size, highly variable in shape, and easily confused with leaf textures, the FADN dynamically fuses high-dimensional semantic features with low-dimensional detail features through an adaptive weighting mechanism to capture subtle morphological differences. When detecting tiny insect bodies, it prioritizes enhancing local details in shallow features. In contrast, under strong background interference, it focuses more on deep features’ global semantic information to suppress noise. This dynamic balancing mechanism enables the model to preserve the fine structures of pest objects while enhancing robustness against complex backgrounds, thereby reducing missed detections and false positives.

### 3.3. Re-Parameterized Batch Normalization

Guo et al. [32] proposed RepBN, an efficient normalization method designed to replace LayerNorm in Transformers by dynamically adjusting the normalization process through combining standard BatchNorm with a learnable parameter η. The core idea is to add a linear term (ηX) of the input X to the output of BatchNorm during training, thereby enhancing the model’s representational capacity. After training, RepBN is mathematically re-parameterized into an equivalent BatchNorm form, allowing it to be directly fused with adjacent linear layers during inference. This eliminates extra computation and reduces latency. To avoid training instability caused by directly replacing LayerNorm, RepBN employs a progressive strategy that gradually reduces dependence on LayerNorm. At the initial stage, the model fully relies on LayerNorm, then progressively transitions to pure BatchNorm as training advances, ensuring stable training, as illustrated in Figure 5.

This design not only retains the efficient inference characteristics of BatchNorm but also improves training stability and flexibility through dynamic parameter adjustment and progressive replacement. It makes the model better suited for resource-constrained deployment environments while maintaining performance. The formula for RepBN is shown in Equation (11) [32]:(11)$$RepBN\left(X\right)=BN\left(X\right)+\eta X$$

Here, BN(X) denotes the standard Batch Normalization operation, and η is a learnable parameter. This design introduces additional flexibility by adding a linear term (ηX) of the input feature X on top of the BatchNorm output, allowing the normalization layer to dynamically adjust the feature distribution. During training, η is optimized via gradients, enhancing the model’s ability to adapt to complex patterns. The re-parameterized form of RepBN is shown in Equation (12) [32]:(12)$$RepBN\left(X\right)=BN\left(X;\mu ,\sigma ,\alpha +\eta \sigma ,\beta +\eta \mu \right)$$

Here, μ and σ are the mean and standard deviation of BatchNorm, while α and β are the original BatchNorm scaling and shifting parameters. After re-parameterization, the new scaling parameter becomes α+ησ, and the new shifting parameter becomes β+ημ. The key to this transformation is incorporating the dynamic adjustment term ηX into the BatchNorm parameters, allowing it to be directly merged with adjacent linear layers during inference without extra computation, thereby improving efficiency.

We transformed the AIFI module of RT-DETR into a RepBN-based architecture by replacing the normalization component of AIFI with RepBN, resulting in AIFI-RepBN. This architecture can be further re-parameterized into BatchNorm and merged with linear layers, significantly improving inference efficiency while maintaining detection accuracy. Its core advantage lies in combining the efficient inference characteristics of BatchNorm with a learnable linear term, enabling the model to reduce extra normalization computations and lower latency when processing high-resolution images and multi-scale objects. At the same time, it adaptively handles complex feature distribution variations, enhancing sensitivity to small pest objects or occluded scenarios. Furthermore, the progressive strategy smoothly transitions from LayerNorm to BatchNorm, avoiding model instability caused by abrupt parameter changes during early training. Additionally, RepBN’s hardware-friendly design makes it easier to deploy on edge devices by simplifying computations through parameter fusion and reducing memory usage, providing an efficient and reliable solution for real-time pest detection in resource-constrained environments.

## 4. Experiment

### 4.1. Experimental Dataset and Environment

The Forestry Pest Dataset [33] was constructed by Liu et al. as a specialized dataset for the identification of pests in economic forestry. It aims to address the low efficiency of manual inspection in traditional methods as well as the small scale and limited diversity of existing datasets. This dataset was constructed through a multi-stage process to ensure quality and reproducibility. Images were collected from search engines and specialized forestry pest websites, covering 31 common forestry pest species across multiple life stages such as eggs, larvae, nymphs, and adults, all captured in natural wild environments. They cover a wide range of ecological groups, including wood borers, defoliators, sap-sucking pests and leaf miners. A preliminary filtering process was carried out by four trained volunteers with the assistance of three forestry experts, who removed duplicate, invalid, or blurred images, resulting in 2278 high-quality original samples. To address class imbalance and improve generalization, seven data augmentation techniques including rotation, noise addition, and brightness transformation were applied, which expanded the dataset to 7163 images. Annotation was performed using the Labelimg tool, where three volunteers labeled pest categories and bounding boxes under the supervision of experts. Ambiguous samples were jointly reviewed and resolved by experts to ensure annotation accuracy. Finally, the dataset was divided into 5730 training images, 716 validation images, and 717 test images. These procedures clearly define the conditions for dataset construction and the criteria for sample selection, thereby enhancing the reproducibility of the experimental results. The 31 pest categories are numbered sequentially from 0 to 30, as shown in Table 1.

The number and size distribution of labels are shown in Figure 6. Figure 6a indicates that the dataset exhibits an imbalance in sample distribution, where different pest categories are distinguished by 31 colors corresponding to class indices from 0 to 30 in the Forestry Pest Dataset. The most frequently occurring categories are “Erthesina fullo (nymph2)” and “Plagiodera versicolora (Laicharting) (ovum)”. These pests often appear in clusters within a single image, resulting in significantly more annotated instances compared to other categories. Similarly, other small-scale objects tend to appear repeatedly in individual images, leading to relatively high occurrence frequencies. This distribution pattern aligns with real-world conditions in economic forestry scenarios. Figure 6b shows that most object aspect ratios in the dataset are concentrated below 0.1, which corresponds to the typical size characteristics of small object detection.

During the data annotation process, forestry experts were invited to participate in classification and labeling to ensure accuracy, using professional tools to assist with the task. The dataset retains the complex morphological variations and life cycle differences among pests in real outdoor environments, such as the developmental changes in the same species at different life stages. Compared to previous forestry pest datasets that were limited to laboratory conditions or focused on a single species, this dataset is more representative of real-world application scenarios. It provides high-quality data support for deep learning research in the monitoring and control of pests in economic forestry. Therefore, we use this dataset for model training. Detailed information about the experimental settings is shown in Table 2.

### 4.2. Evaluation Metrics

In object detection tasks, Precision, Recall, and mean Average Precision (mAP) are the most commonly used evaluation metrics. Precision measures the proportion of correctly identified targets among all the targets predicted by the model, as shown in Equation (13). A high precision indicates that most of the detected targets are accurate and reliable, while a low precision suggests a high number of false positives. Recall represents the proportion of correctly detected true targets out of all actual targets, as shown in Equation (14). A high recall indicates that the model successfully captures the majority of targets, while a low recall suggests a high rate of missed detections.(13)$$Precision=\frac{TP}{TP+FP}$$(14)$$Recall=\frac{TP}{TP+FN}$$

Here, TP (True Positive) refers to the number of correctly detected targets, FP (False Positive) is the number of incorrectly detected targets, and FN (False Negative) is the number of targets that were not correctly detected.

Average Precision (AP) evaluates the detection accuracy of a single class by computing the average precision across different recall levels. Mean Average Precision (mAP) is then calculated by averaging the AP values of all classes, providing a comprehensive assessment of the model’s overall detection performance, as shown in Equations (15) and (16).(15)$$AP=\int_{0}^{1}P\left(r\right)dr$$(16)$$mAP=\frac{1}{C}\sum_{i=1}^{C}{AP}_{i}$$

In the formulas, P(r) represents the precision at a given recall rate r, and C denotes the total number of classes.

${mAP}_{0.5}$ refers to the mean Average Precision calculated at a specific threshold (IoU = 0.5), where IoU (Intersection over Union) measures the overlap between the predicted bounding box and the ground truth box. This metric evaluates the model’s detection accuracy under a relatively lenient IoU requirement. In contrast, ${mAP}_{0.5:0.95}$ is the mean Average Precision computed across a range of IoU thresholds from 0.5 to 0.95 with a step size of 0.05. Compared to ${mAP}_{0.5}$, ${mAP}_{0.5:0.95}$ provides a more rigorous evaluation, as it requires the model to maintain high localization accuracy under stricter IoU conditions.

FPS (frames per second) is a key metric for evaluating a model’s processing efficiency, directly reflecting the algorithm’s ability to analyze image data within a unit of time. An increase in this parameter corresponds to enhanced real-time processing performance, meaning the system can achieve faster visual information recognition and response in time-sensitive scenarios.

Additionally, the number of model parameters is generally related to the model’s complexity. A larger number of parameters may indicate stronger fitting capability but also increases computational cost and storage requirements. The goal of lightweight models is to reduce the number of parameters while maintaining high performance. FLOPs (floating point operations) refer to the total number of floating-point calculations performed during model inference, usually measured in GFLOPs (billion floating-point operations), and are used to assess the computational complexity of a model. A higher FLOPs value means more inference time and hardware resources are required. In object detection tasks, an ideal model should achieve a high ${mAP}_{0.5:0.95}$ while minimizing both the number of parameters and FLOPs to meet deployment demands in practical scenarios. Therefore, this paper selects ${mAP}_{0.5:0.95}$, model parameter count, and FLOPs as the evaluation metrics.

### 4.3. Ablation Experiments

To verify the effectiveness of each improved module proposed in the LightFAD-DETR model, ablation experiments are conducted in this section. In comparison with the baseline model, the following components are individually integrated into the RT-DETR framework: replacing the backbone with the first nine layers of YOLOv9, incorporating the RepNCSPELAN4-CAA module, deploying the FADN with the DASI module, and replacing the AIFI module with AIFI-RepBN. Subsequently, all four improvements are progressively integrated to form the complete LightFAD-DETR model.

When only replacing the backbone with the first nine layers of YOLOv9, the ${mAP}_{0.5:0.95}$ drops by 0.2% compared to the baseline model, but the number of parameters decreases by 47.2% and the computational cost is reduced by 40.8%, making the model more lightweight. When only the RepNCSPELAN4-CAA module is added by replacing the RepC3 module in the RT-DETR model, the ${mAP}_{0.5:0.95}$ increases by 0.5 percent, enhancing the semantic perception capability of feature fusion, while the parameter count and computational load also decrease to some extent. When only the FADN with the DASI module is integrated, the ${mAP}_{0.5:0.95}$ increases by 1.5%, achieving efficient collaborative modeling of details and semantics in complex backgrounds, although this leads to some increase in both parameters and computation. When the AIFI-RepBN module is added alone, the ${mAP}_{0.5:0.95}$ improves by 0.8% without significantly changing the model’s parameter size or computational load, greatly enhancing inference efficiency while maintaining detection accuracy. When the YOLOv9 backbone is used in combination with replacing both the RepNCSPELAN4 modules in the backbone and the RepC3 module in the model with the RepNCSPELAN4-CAA modules, the model’s parameters and computational cost are further reduced, and ${mAP}_{0.5:0.95}$ increases by 0.3%. Following this, the integration of the DASI-based FADN brings a 0.7% increase in ${mAP}_{0.5:0.95}$, with only an 11.5% increase in parameters and a 14.2% increase in FLOPs. Finally, by incorporating the AIFI-RepBN module, the model achieves an additional 0.6% increase in ${mAP}_{0.5:0.95}$ without significantly affecting parameters or computation. In summary, compared to the baseline, the final LightFAD-DETR model achieves a 1.4% improvement in ${mAP}_{0.5:0.95}$, with a 41.7% reduction in parameters and a 35.0% reduction in computational cost, thus achieving a dual improvement in both accuracy and lightweight performance. Detailed results are shown in Table 3.

Vaswani et al. [34] proposed the Learned Positional Encoding method, which was used to enhance the positional encoding mechanism of the AIFI module, resulting in the AIFI-LPE module. Shaker et al. [35], in the SwiftFormer model, introduced the Efficient Additive Attention mechanism, which was applied to improve the global attention mechanism of the AIFI module, leading to the AIFI-EfficientAdditive module. Xia et al. [36] proposed Deformable Attention, which was incorporated into the AIFI module’s global self-attention mechanism, resulting in the AIFI-DAttention module. Pan et al. [37] introduced the HiLo Attention mechanism that captures both high and low frequency information and was used to improve the attention mechanism in AIFI, resulting in the AIFI-HiLo module. Wu et al. [38] proposed the M2SA module, which includes two branches: the first employs an enhanced channel attention mechanism, while the second integrates multi-scale depthwise separable dilated convolutions and adaptive pooling in the MSMHSA (Multi-Scale Multi-Head Self-Attention) module. We integrated the MSMHSA module into the AIFI structure, forming the AIFI-MSMHSA module.

We integrated different improved versions of the AIFI module into the RT-DETR model to evaluate the suitability of the AIFI-RepBN module. As shown in Table 4, all modified modules maintain nearly the same parameter count and computational cost. The inclusion of the AIFI-LPE and AIFI-EfficientAdditive modules resulted in a slight decrease in the original model’s ${mAP}_{0.5:0.95}$. The AIFI-DAttention and AIFI-HiLo modules had minimal impact on the ${mAP}_{0.5:0.95}$. In contrast, both the AIFI-RepBN and AIFI-MSMHSA modules led to a noticeable improvement in the model’s ${mAP}_{0.5:0.95}$.

To demonstrate that the AIFI-RepBN module is more suitable for the LightFAD-DETR model, we integrated six different improved AIFI modules into the LightFAD-DETR model separately. The results are shown in Table 5. All improved models have similar parameter counts and computational costs. Among them, the AIFI-RepBN, AIFI-DAttention, and AIFI-HiLo modules all achieved improvements in ${mAP}_{0.5:0.95}$ based on the LightFAD-DETR model. Notably, the integration of the AIFI-RepBN module led to the most significant improvement in ${mAP}_{0.5:0.95}$ while maintaining the lowest computational cost, making it the most suitable for the LightFAD-DETR model. Therefore, we adopt the AIFI-RepBN module in the final model.

In the experiment regarding improving the RepC3 module in the baseline model, we integrated different improved versions of the RepC3 module into the RT-DETR model to verify the suitability of the RepNCSPELAN4-CAA module. Ding et al. [39] proposed the Diverse Branch Block, which can convert complex multi-branch designs into a single convolutional layer using structural re-parameterization techniques. This allows the model to maintain high expressive power during training and lightweight structure during inference. We introduced this block by replacing the RepBlock in the RepC3 module, forming the DBBC3 module. Gong [40] proposed the Dynamic Group Shuffle Transformer (DGST), which uses a 3:1 splitting strategy to combine group convolution with channel shuffle operations, and replaces traditional linear layers with convolutional layers. This design reduces computational complexity while enhancing the model’s ability to capture both local and global features. In this study, we replaced the RepC3 module with the DGST module to conduct comparative experiments. Song et al. [41] introduced the gConv Block, which combines gating mechanisms with depthwise separable convolutions to enhance model expressiveness while maintaining computational efficiency. We used this block to replace the RepBlock in the RepC3 module, resulting in the gConvC3 module. Ding et al. [42] also proposed the Dilated Reparam Block in UniRepLKNet, which allows flexible adjustment of dilation rates and kernel sizes to dynamically adapt to different receptive field and detail extraction requirements. We used this block to replace the RepBlock in the RepC3 module, forming the DRBC3 module. Finally, we replaced the RepC3 module with the RepNCSPELAN4 module from YOLOv9 and conducted comparative experiments with the other improved versions. As shown in Table 6, all the improved modules reduced model parameters and computational cost to varying degrees. However, only the DBBC3 module and the RepNCSPELAN4-CAA module improved the model’s ${mAP}_{0.5:0.95}$, with the RepNCSPELAN4-CAA module achieving better improvement while maintaining lower parameter count and computation cost.

To demonstrate that the RepNCSPELAN4-CAA module is more suitable for the LightFAD-DETR model, we integrated six different improved versions of the RepC3 module into the LightFAD-DETR model. The results are shown in Table 7. Among all the improvements, the DBBC3 module and the RepNCSPELAN4-CAA module showed the most significant improvement in ${mAP}_{0.5:0.95}$, with the RepNCSPELAN4-CAA module achieving the best performance. However, the ${mAP}_{0.5:0.95}$ improvement from the DBBC3 module came at the cost of increased model parameters and computational complexity. Although the other improved modules further reduced the model’s complexity, their ${mAP}_{0.5:0.95}$ performance was not satisfactory. Therefore, we incorporated the RepNCSPELAN4-CAA module into the LightFAD-DETR model.

### 4.4. Classification Performance Analysis

To evaluate the overall performance of the model, we conducted a series of comparative experiments between the LightFAD-DETR model and RT-DETRr18, RT-DETRr34, and RT-DETRr50. As shown in Table 8, the LightFAD-DETR model outperforms all versions of RT-DETR in terms of accuracy and ${mAP}_{0.5:0.95}$. In terms of model size and computational complexity, LightFAD-DETR has about one-third the parameters and FLOPs of RT-DETRr34, and about one-fourth of RT-DETRr50. Moreover, its FPS surpasses both RT-DETRr34 and RT-DETRr50, significantly exceeding the real-time detection requirements for forestry pest monitoring.

Figure 7 presents a comparison of the ${mAP}_{0.5:0.95}$ across 31 categories between the LightFAD-DETR model and the RT-DETRr18 model. The results show that the LightFAD-DETR model achieves higher detection accuracy in approximately two-thirds of the categories, while the difference in accuracy for the remaining less favorable categories is relatively small. Notably, significant improvements in ${mAP}_{0.5:0.95}$ are observed for categories such as “Drosicha contrahens (female)”, “Erthesina fullo”, “Cnidocampa flavescens”, “Plagiodera versicolora (Laicharting) (larvae)”, “Plagiodera versicolora (Laicharting) (ovum)”, “Apriona germari (Hope)”, and “Latoia consocia (Walker)”. These improvements highlight the LightFAD-DETR model’s strong ability to capture local detail information, its advantage in detecting small objects, and its robustness against background interference.

### 4.5. Performance Comparison of Different Models

Table 9 presents the comparison results between the LightFAD-DETR model and several popular object detection models. The comparison includes two-stage detectors (Faster R-CNN [43]), one-stage detectors (SSD512 [44], RetinaNet [45], and the YOLO series), and Transformer-based detectors (DETR [46], Deformable DETR [47], Swin-B [48]). The analysis shows that the LightFAD-DETR model achieves an outstanding balance between accuracy, lightweight design, and speed, significantly outperforming other models. In terms of detection accuracy, the LightFAD-DETR model reaches a ${mAP}_{0.5:0.95}$ of 88.5% and a precision of 97.3%, placing it at a top-tier level. Its performance is comparable to or even better than YOLOv8m and clearly surpasses other lightweight models such as YOLOv5m and YOLOv10m [49], as well as classic models like Faster R-CNN and SSD512.

The core advantage of the LightFAD-DETR model lies in its exceptionally lightweight design and efficiency, all while maintaining high accuracy. It has the lowest computational complexity and number of parameters among all the compared models, significantly lower than models such as Faster R-CNN, DETR, and Swin-B. Despite this substantial reduction in model size, it does not compromise on speed, as it achieves an inference speed of 106.3 FPS, which far exceeds real-time detection requirements. Although slightly lower than the fastest models, YOLOv8m (137.8 FPS) and YOLOv10m (142.5 FPS), the LightFAD-DETR model achieves nearly a 50% reduction in parameters and FLOPs while offering comparable or even superior accuracy. Therefore, LightFAD-DETR is a lightweight model that maintains high accuracy and speed, making it well-suited for deployment on edge devices under the strict resource constraints and real-time demands of intelligent forestry pest detection.

### 4.6. Performance Analysis on Different Datasets

To demonstrate the generalization capability and deployment advantages of the LightFAD-DETR model in complex, multi-category pest scenarios, we conducted comparative experiments on the IP102 dataset [52]. The IP102 dataset provides strong data support for pest detection in economic forest settings, covering 102 common pest species and meeting the specific identification needs of such environments. With over 75,000 images captured in real-world conditions, the dataset comprehensively represents pests across different developmental stages and includes complex background interference, effectively enhancing the model’s generalization ability in diverse forestry scenarios. In addition, approximately 19,000 images with bounding box annotations are available for training object detection models, enabling integrated pest localization and classification analysis.

The detection performance comparison of common training models is shown in Table 10. The results indicate that all models experience a decline in detection accuracy on this dataset compared to the Forestry Pest Dataset. However, the LightFAD-DETR model remains in the top tier, achieving a ${mAP}_{0.5:0.95}$ of 41.2% with extremely low resource consumption. Its accuracy is comparable to mainstream models such as YOLOv8m and RT-DETRr50, and it significantly outperforms traditional detectors like Faster R-CNN and SSD512. Furthermore, its high inference speed of 106.3 FPS surpasses most Transformer-based models, fully meeting real-time application demands. These results highlight the effectiveness of LightFAD-DETR’s lightweight design and demonstrate its strong performance in complex, cross-dataset pest detection tasks. The model maintains high accuracy and stability when facing new categories and unfamiliar backgrounds, effectively establishing strong discriminative feature representations for diverse target morphologies. It achieves an optimal balance of speed, accuracy, and efficiency under edge device resource constraints, providing a reliable and high-performance foundation for intelligent pest detection in economic forestry applications.

### 4.7. Visual Analysis of Detection Results

Figure 8 presents a visual comparison of pest target detection results between the RT-DETR model and the proposed LightFAD-DETR model on the Forestry Pest Dataset. In each group, the left column shows the original image, the middle column displays the detection results from the RT-DETR model, and the right column shows the results from the LightFAD-DETR model. Each row represents an independent comparison group, highlighting the performance differences between the two models in challenging scenarios. In the first row, the RT-DETR model shows clear limitations when dealing with blurred images and densely packed objects, failing to detect a large number of pest eggs. In contrast, the LightFAD-DETR model successfully identifies all targets in the scene, overcoming the difficulties of blur and occlusion, and generally assigns higher confidence scores to the detected objects. In the second row, the RT-DETR model incorrectly identifies parts of the image background as pest eggs, which could lead to unnecessary alerts or interventions in real-world applications. The LightFAD-DETR model, however, avoids these false detections. In the third row, at the bottom right of the image, the RT-DETR model mistakenly detects two closely adjacent pests as a single target, while the LightFAD-DETR model accurately distinguishes and detects both targets correctly. In summary, through these three representative comparison scenarios, the LightFAD-DETR model demonstrates significant advantages over the RT-DETR model in handling image blur, dense object distributions, background interference, and closely located objects. These results strongly validate the superior accuracy and robustness of the LightFAD-DETR model for pest detection tasks in economic forestry.

## 5. Discussion

Through its innovative lightweight architecture and effective multi-module collaborative optimization, LightFAD-DETR successfully achieves an excellent balance of accuracy, speed, and model efficiency in the task of economic forestry pest detection. Its strong robustness in complex backgrounds, effective detection of small and slender objects, and demonstrated generalization ability across datasets fully validate the model’s advanced design and practical applicability. LightFAD-DETR provides strong technical support for the edge deployment of intelligent forestry pest-monitoring systems and holds significant application value and broad prospects for adoption.

Compared with previously known pest detection models [43,44,45,46,47,48,49,50,51], LightFAD-DETR achieves a more balanced trade-off between accuracy, efficiency, and deployment feasibility. Traditional approaches often emphasize accuracy at the expense of model size and inference time, making them difficult to use in resource-constrained forestry environments. In contrast, our model integrates lightweight architectural design with enhanced feature aggregation, thereby preserving fine-grained details of small pest targets while maintaining high processing speed. From a scientific perspective, this work contributes a novel detection paradigm that demonstrates how advanced feature modeling strategies and re-parameterization techniques can be effectively combined in economic forest pest monitoring, addressing challenges such as complex backgrounds and small object recognition. From a practical perspective, LightFAD-DETR offers a deployable solution for intelligent forestry management, enabling real-time, large-scale, and cost-effective pest monitoring in real-world scenarios. These advantages underscore both the theoretical significance and the applied value of the model, establishing it as a step forward toward bridging the gap between laboratory research and field application.

In addition to experimental validation, the LightFAD-DETR model was developed with practical deployment in mind. Its lightweight structure and reduced computational requirements make it well suited for embedding into edge devices such as portable monitoring instruments, mobile phones, or compact computing units with GPU or NPU acceleration. This enables real-time pest detection directly in the field, even in remote forestry areas with limited network connectivity, thereby reducing latency and improving system reliability. The model can also be integrated into unmanned aerial vehicles for large-scale aerial surveys or combined with fixed ground-based monitoring stations and Internet of Things platforms to establish continuous and automated pest surveillance systems. These application pathways highlight the feasibility of deploying LightFAD-DETR in real-life forestry scenarios, bridging the gap between academic research and practical forest pest management.

This study’s analysis is entirely based on publicly available datasets. The model’s performance potential is constrained by the quality and inherent limitations of these datasets, which in turn affects the robustness of the study’s conclusions. In terms of internal validity, a key challenge lies in the quality of the datasets themselves. For instance, inadequate sampling strategies may fail to comprehensively cover the object phenomena, and subjective inconsistencies during data annotation may introduce potential biases into the model’s learning process, leading to distorted findings. From the perspective of external validity, the limitations of the datasets pose major constraints, such as sample size, coverage scope, and characteristics of the specific data collection environment. These factors collectively may weaken the generalizability of the study’s conclusions to broader contexts. The observed accuracy decline on the IP102 dataset suggests that there is still room for improvement when the model is faced with larger volumes of data, finer objects, or extremely dense and overlapping instances. The recognition of tiny pests under multilayer leaf occlusion also remains a persistent challenge. Although the model accounts for different developmental stages of pests, its robustness in recognizing species with highly variable morphologies, or those in non-standard postures or damaged conditions, requires further validation and enhancement.

Future research will further enhance the adaptability of the LightFAD-DETR model in complex forestry environments. To address the challenges of extremely small objects and heavy occlusion, we plan to integrate super-resolution techniques and adaptive receptive field mechanisms to improve the model’s ability to detect tiny pests hidden beneath deep leaf coverage. Additionally, to reduce the model’s reliance on annotated data, we aim to explore its integration with few-shot learning and transfer learning strategies. On the lightweight design front, we will investigate dynamic neural architecture search (NAS) techniques to generate customized subnetworks tailored to the hardware characteristics of different edge devices, achieving a refined balance between accuracy and efficiency.

## 6. Conclusions

This paper proposes a lightweight pest detection model, LightFAD-DETR, designed for complex economic forestry scenarios. By implementing multidimensional collaborative optimization based on the RT-DETR framework, the model achieves an efficient balance between accuracy, speed, and resource consumption. First, the introduction of the YOLOv9 backbone significantly reduces the computational load. Then, the RepNCSPELAN4-CAA module is proposed, which integrates re-parameterization techniques with one-dimensional strip convolutions to enhance cross-region modeling capabilities for elongated pest objects while reducing inference latency. Additionally, a feature aggregation diffusion network is designed, along with the introduction of a dimension-aware selective integration module. These components dynamically fuse deep and shallow features, effectively mitigating information loss for small objects in complex backgrounds. Furthermore, the AIFI module, improved with re-parameterized batch normalization and trained using a progressive training strategy, eliminates redundant computation and further enhances deployment efficiency on edge devices. Experimental results demonstrate that, while significantly reducing parameter count and computational complexity, the proposed model surpasses the baseline in detection accuracy and meets real-time inference requirements. In cross-dataset evaluations, it exhibits generalization performance comparable to mainstream models, particularly excelling in handling blurred images, dense objects, and complex background interference, effectively reducing missed and false detections. In terms of quantitative performance, LightFAD-DETR achieved a mean average precision of 88.5% (${mAP}_{0.5:0.95}$) on the Forestry Pest Dataset, surpassing existing baseline models by a clear margin in both overall accuracy and small object recognition. The model runs at 106.3 FPS on RTX 2080 GPU, with only 11.6 million parameters and 31.7 GFLOPs, demonstrating its ability to maintain real-time performance while remaining lightweight. Compared with previous studies, this model achieves breakthrough advancements in both lightweight design and real-time performance, offering a highly efficient and reliable solution for edge deployment of intelligent pest detection terminals in economic forestry.

## Figures and Tables

**Figure 1 insects-16-00959-f001:**
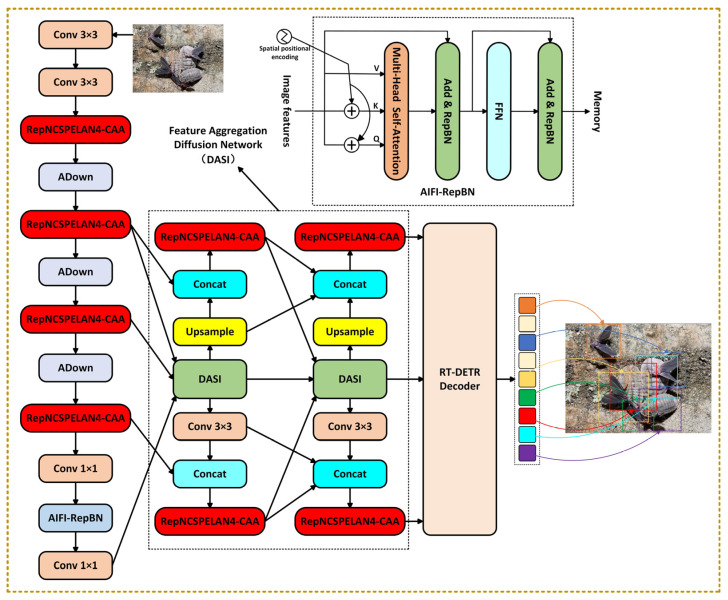
The network structure of the LightFAD-DETR model.

**Figure 2 insects-16-00959-f002:**
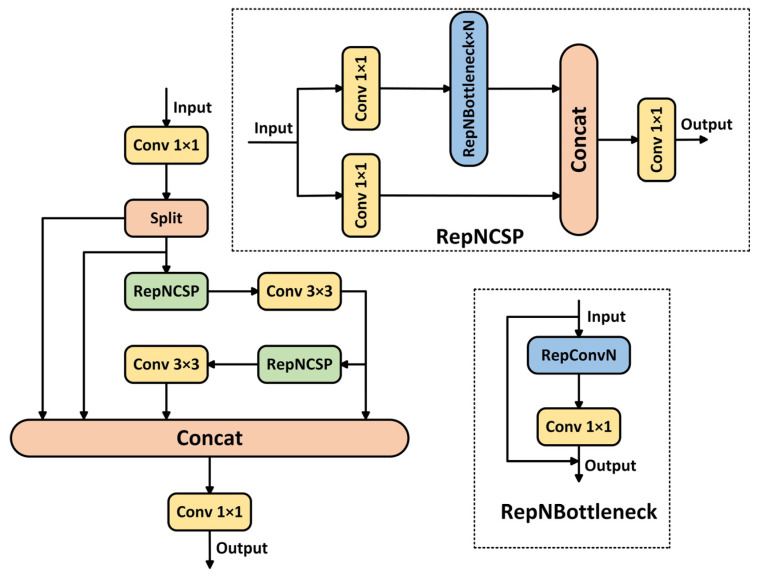
Module structure of RepNCSPELAN4.

**Figure 3 insects-16-00959-f003:**
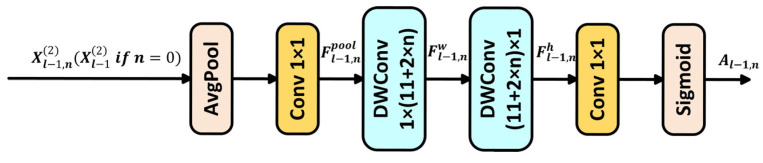
Context anchor attention module.

**Figure 4 insects-16-00959-f004:**
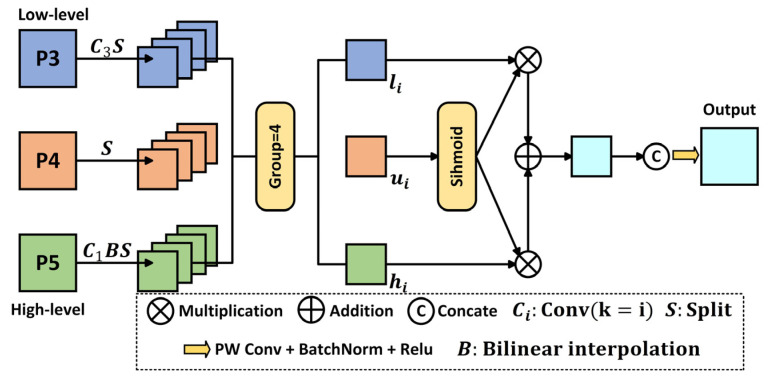
Dimension-Aware Selective Integration module.

**Figure 5 insects-16-00959-f005:**
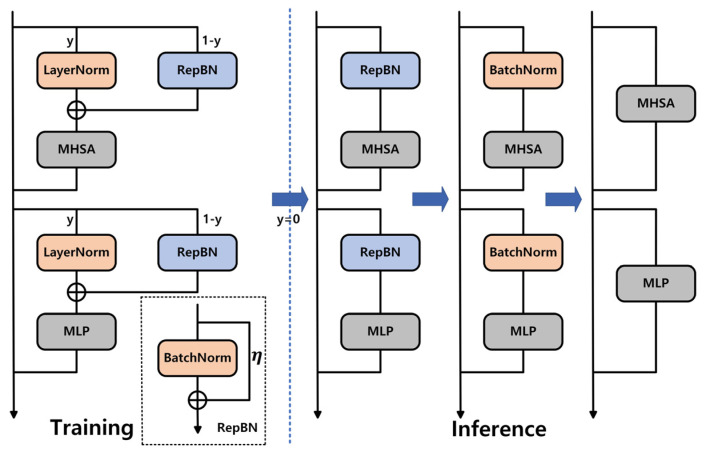
The progressive strategy of RepBN.

**Figure 6 insects-16-00959-f006:**
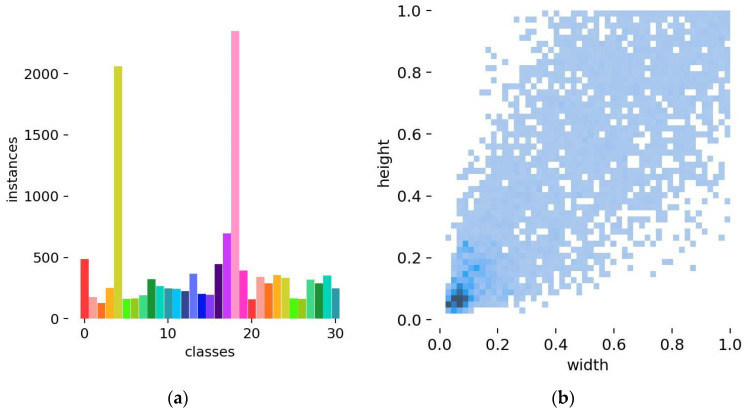
Tag quantity map (**a**) and label distribution size map (**b**).

**Figure 7 insects-16-00959-f007:**
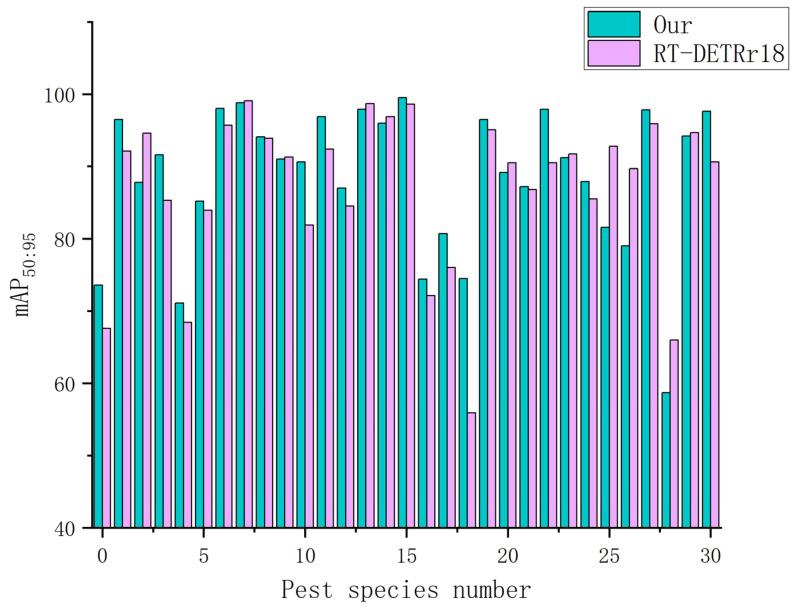
Comparison of ${mAP}_{0.5:0.95}$ between RT-DETR and LightFAD-DETR models in each pest category.

**Figure 8 insects-16-00959-f008:**
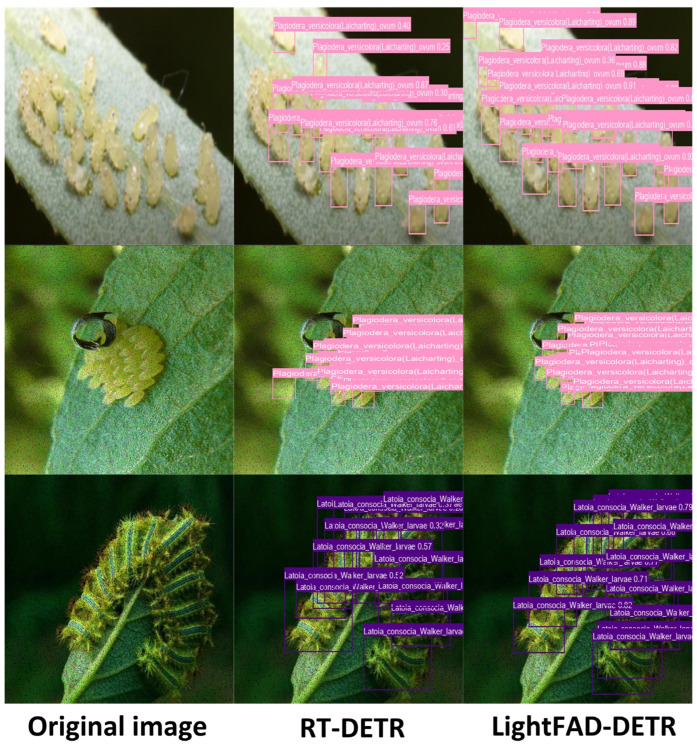
Comparison of detection results between the RT-DETR and the LightFAD-DETR.

**Table 1 insects-16-00959-t001:** The image sample numbers of 31 types of pests in the Forest Pest Dataset.

Label Number	Category	Label Number	Category
0	Drosicha contrahens (female)	16	Latoia consocia (Walker) (larvae)
1	Drosicha contrahens (male)	17	Plagiodera versicolora (Laicharting) (larvae)
2	Chalcophora japonica	18	Plagiodera versicolora (Laicharting) (ovum)
3	Erthesina fullo	19	Plagiodera versicolora (Laicharting)
4	Erthesina fullo (nymph2)	20	Spilarctia subcarnea (Walker) (larvae)
5	Erthesina fullo (nymph)	21	Spilarctia subcarnea (Walker) (larvae2)
6	Spilarctia subcarnea (Walker)	22	Apriona germari (Hope)
7	Psilogramma menephron	23	Hyphantria cunea
8	Sericinus montela	24	Cerambycidae (larvae)
9	Sericinus montela (larvae)	25	Psilogramma menephron (larvae)
10	Clostera anachoreta	26	Monochamus alternatus
11	Cnidocampa flavescens (Walker)	27	Micromelalopha troglodyte (Graeser) (larvae)
12	Cnidocampa flavescens (Walker) (pupa)	28	Hyphantria cunea (larvae)
13	Anoplophora chinensis	29	Hyphantria cunea (pupa)
14	Micromelalopha troglodyte (Graeser)	30	Latoia consocia (Walker)
15	Psacothea hilari (Pascoe)		

**Table 2 insects-16-00959-t002:** Experimental configuration details, including hardware environment, software framework, and training parameter settings.

Experimental Configuration	Configuration Information
Deep Learning Framework	Pytorch 2.0.0
Software Version	PyCharm PROFESSIONAL 2023.1
Python Version	Python 3.8.19
Server	AMAX
GPU	NVIDIA GeForce 2080 Ti
Number of Training Epochs	300
Training Batch Size	16
Number of Threads	4
Optimizer	0.01

**Table 3 insects-16-00959-t003:** Ablation experiments compared with the baseline model.

Baseline Model	Replace the Backbone	Rep-NCSPE-LAN4-CAA	FADN (DASI)	AIFI-RepBN	${mAP}_{0.5:0.95}$ (%)	Params (M)	GFLOPs
RT-DETR					87.1	19.9	57.1
	√				86.9	10.5	33.8
		√			87.6	19.1	53.1
			√		88.6	21.2	63.5
				√	87.9	19.9	57.1
	√	√			87.2	10.4	32.5
	√	√	√		87.9	11.6	37.1
	√	√	√	√	88.5	11.6	37.1

**Table 4 insects-16-00959-t004:** The experiment regarding improving the AIFI module in the baseline model.

Improved AIFI Module	${mAP}_{0.5:0.95}$(%)	Params (M)	GFLOPs
AIFI-LPE	86.4	20.0	57.1
AIFI-EfficientAdditive	86.6	19.9	57.3
AIFI-DAttention	87.2	19.9	57.3
AIFI-HiLo	87.1	19.9	57.2
AIFI-MSMHSA	87.6	19.9	57.2
AIFI-RepBN	**87.9**	**19.9**	**57.1**

Bold text represents prominent advantageous data.

**Table 5 insects-16-00959-t005:** The experiment regarding improving the AIFI module in the LightFAD-DETR model.

Improved AIFI Module	${mAP}_{0.5:0.95}$(%)	Params (M)	GFLOPs
AIFI-LPE	87.9	11.8	37.3
AIFI-EfficientAdditive	86.9	11.7	37.6
AIFI-DAttention	88.0	11.6	37.6
AIFI-HiLo	88.1	11.6	37.5
AIFI-MSMHSA	87.6	11.7	37.5
AIFI-RepBN	**88.5**	**11.6**	**37.1**

Bold text represents prominent advantageous data.

**Table 6 insects-16-00959-t006:** The experiment regarding replacing the RepC3 module in the RT-DETR model.

Replace the RepC3 Module	${mAP}_{0.5:0.95}$(%)	Params (M)	GFLOPs
DBBC3	87.5	19.9	57.1
DGST	87.1	18.6	50.4
gConvC3	84.9	18.6	51.2
DRBC3	86.5	18.2	48.4
RepNCSPELAN4	86.7	18.6	50.3
RepNCSPELAN4-CAA	**87.6**	19.1	53.1

Bold text represents prominent advantageous data.

**Table 7 insects-16-00959-t007:** The experiment regarding replacing the RepC3 module in the LightFAD-DETR model.

Replace the RepC3 Module	${mAP}_{0.5:0.95}$(%)	Params (M)	GFLOPs
DBBC3	88.1	12.6	43.1
DGST	87.4	11.2	34.0
gConvC3	86.1	11.4	35.4
DRBC3	87.3	10.7	31.6
RepNCSPELAN4	87.2	11.1	33.6
RepNCSPELAN4-CAA	**88.5**	11.6	37.1

Bold text represents prominent advantageous data.

**Table 8 insects-16-00959-t008:** Comparison of performance indicators between the LightFAD-DETR model and the RT-DETR model.

Model	Precision (%)	Recall (%)	${mAP}_{0.5:0.95}$ (%)	Params (M)	GFLOPs	FPS (Frame/s)
RT-DETRr18	96.3	93.9	87.1	19.9	57.1	141.5
RT-DETRr34	96.6	95.3	87.9	31.2	88.9	103.9
RT-DETRr50	97.0	95.6	87.1	42.0	129.7	62.5
LightFAD-DETR	**97.3**	94.8	**88.5**	**11.6**	**31.7**	106.3

Bold text represents prominent advantageous data.

**Table 9 insects-16-00959-t009:** Comparison of performance indicators in different training models of the LightFAD-DETR model.

Model	Precision (%)	Recall (%)	${mAP}_{0.5:0.95}$ (%)	Params (M)	GFLOPs	FPS (Frame/s)
Faster R-CNN	90.2	86.7	77.4	137.1	361.6	13.6
SSD512	92.1	93.6	81.6	24.8	97.5	42.1
RetinaNet	96.1	93.8	86.4	36.5	198.2	24.7
YOLOv5m	95.4	94.2	87.3	41.2	63.7	102.4
YOLOv8m	97.8	95.0	88.3	23.2	67.5	137.8
YOLOv10m	96.7	93.5	86.9	15.7	59.1	142.5
YOLO11m [50]	97.1	94.8	87.7	20.3	67.9	113.6
YOLOv12m [51]	96.8	95.4	87.9	20.1	67.7	103.2
DETR	95.6	93.9	86.6	41.1	86.0	14.2
Deformable DETR	95.8	94.3	87.1	39.4	173.2	22.9
Swin-B	96.7	94.5	87.7	88.3	47.0	11.4
LightFAD-DETR	97.3	94.8	**88.5**	**11.6**	**31.7**	106.3

Bold text represents prominent advantageous data.

**Table 10 insects-16-00959-t010:** Comparison of performance indicators between the LightFAD-DETR model and common training models on the IP102 dataset.

Model	Precision (%)	Recall (%)	${mAP}_{0.5:0.95}$ (%)	Params (M)	GFLOPs	FPS (Frame/s)
Faster R-CNN	44.3	42.9	28.6	137.1	361.6	13.6
SSD512	52.1	48.3	32.1	24.8	97.5	42.1
RetinaNet	62.7	58.6	39.9	36.5	198.2	24.7
YOLOv5m	65.7	60.8	40.9	41.2	63.7	102.4
YOLOv8m	66.9	62.2	41.2	23.2	67.5	137.8
DETR	59.0	54.3	38.4	41.1	86.0	14.2
Deformable DETR	64.5	57.9	39.7	39.4	173.2	22.9
Swin-B	63.2	61.4	40.5	88.3	47.0	11.4
RT-DETRr18	65.2	59.7	40.1	19.9	57.1	141.5
RT-DETRr34	67.1	61.6	40.7	31.2	88.9	103.9
RT-DETRr50	66.7	62.3	41.2	42.0	129.7	62.5
LightFAD-DETR	66.4	61.2	**41.2**	**11.6**	**31.7**	106.3

Bold text represents prominent advantageous data.

## Data Availability

The raw data supporting the conclusions of this article will be made available by the authors on request.

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
