# Peer review of "Lightweight Pest Object Detection Model for Complex Economic Forest Tree Scenarios"

_insects, 2025, doi:10.3390/insects16090959_

Round 1
Reviewer 1 Report
Comments and Suggestions for Authors
The reviewed article is devoted to the current direction of information technology development, which consists in the development of a pest detection model. The authors applied relevant and effective technologies in their research. The article is well-structured and has scientific and practical significance. However, there are some comments on the article that the authors should consider and correct. The main comments are as follows:
- In the introduction section, the motivation for the research should be stated more clearly by identifying the main shortcomings and limitations of known solutions. What is the main planned scientific novelty of the research in the article?
- In the second section, it would be advisable to provide information on the technical means used to deploy the known models.
- In subsection 3.1, it is necessary to describe in more detail the conditions for obtaining the dataset, as well as the criteria used to select the samples. This will increase the reproducibility of the research results obtained.
- It would be advisable to supplement the research results presented in subsections 4.3–4.6 with graphical interpretations, which would improve the perception and analysis of the article's materials by readers.
- The text labels in Figure 8 are illegible. The quality of this figure needs to be improved.
- It is necessary to provide information on the means by which the developed model is planned to be deployed. How can it be applied in real-life scenarios?
- In the discussion section, it is necessary to pay more attention to comparing the developed model with previously known ones. What is the scientific and practical value of the obtained model? This should be clearly stated.
- It is advisable to indicate the main quantitative characteristics of the developed model in the conclusions.
- References to sources from which calculation formulas have been taken must be indicated in the text of the article. Alternatively, authors must indicate that these formulas are their own.
- Figure 6a: Authors should add a description of the colour labels in this figure.
Author Response
Comments 1:
In the introduction section, the motivation for the research should be stated more clearly by identifying the main shortcomings and limitations of known solutions. What is the main planned scientific novelty of the research in the article?
Response 1:
Thank you for pointing this out. We agree with this comment. Therefore, we have revised the Introduction to explicitly identify the main shortcomings and limitations of existing approaches, such as difficulties in detecting tiny pests, weak robustness in complex forest backgrounds, and the insufficient lightweight design of many models for deployment on edge devices. We have also added a clear statement of the planned scientific novelty of our study, emphasizing that the proposed LightFAD-DETR combines lightweight architectural design, enhanced cross-regional feature modeling, and adaptive small-object representation while maintaining real-time inference efficiency.
This revision can be found in the revised manuscript on page 3, lines 97–116, where the new text has been highlighted in red.
Comments 2:
In the second section, it would be advisable to provide information on the technical means used to deploy the known models.
Response 2:
Agree. We have, accordingly, revised Section 2 to include information on the technical means used to deploy previously known models. Specifically, we added descriptions of how some studies integrated their models into IoT-based monitoring systems or optimized them for edge-device deployment, while also noting that most existing works remain primarily theoretical. These additions can be found in the revised manuscript on page 4, lines 154-157 and lines 178-181, with the new text marked in red.
Comments 3:
In subsection 3.1, it is necessary to describe in more detail the conditions for obtaining the dataset, as well as the criteria used to select the samples. This will increase the reproducibility of the research results obtained.
Response 3:
Thank you for this helpful suggestion. We agree with the reviewer’s comment. We assume the reviewer refers to Subsection 4.1 (Experimental Dataset and Environment). In this section, we have revised the text to provide a more detailed description of the conditions under which the dataset was obtained, as well as the criteria applied in selecting the samples. Specifically, we clarified the sources of the images, the filtering process, the expert involvement, and the augmentation methods applied, so as to enhance reproducibility for future studies.
This revision can be found in the revised manuscript on page 12, lines 451–467, with the new text marked in red.
Comments 4:
It would be advisable to supplement the research results presented in subsections 4.3–4.6 with graphical interpretations, which would improve the perception and analysis of the article's materials by readers.
Response 4:
Thank you for this valuable suggestion. We agree that graphical interpretations are often useful for improving data perception. However, in the current study, the results in Subsections 4.3–4.6 involve detailed quantitative comparisons across multiple models, metrics, and ablation settings. We believe that presenting these results in tabular form provides a clearer and more precise representation of the specific numerical differences, which may be less evident in graphical form. Therefore, we have retained the tables to ensure that readers can directly compare exact values.
Comments 5:
The text labels in Figure 8 are illegible. The quality of this figure needs to be improved.
Response 5:
Thank you for this valuable suggestion. We agree with the reviewer’s comment. Accordingly, we have replaced Figure 8 with a clearer version to improve the legibility of the labels and overall figure quality. It should be noted that the original dataset itself contains some relatively low-resolution or blurred images, as it was constructed to reflect realistic and diverse forestry monitoring conditions. Nevertheless, in the revised Figure 8, the detection results and bounding boxes produced by our model are clear and interpretable, which highlights the model’s ability to identify pests even under challenging image conditions.
Comments 6:
It is necessary to provide information on the means by which the developed model is planned to be deployed. How can it be applied in real-life scenarios?
Response 6:
Thank you for this helpful comment. We agree with the reviewer’s suggestion and have revised the Discussion section to add information on how the proposed model can be deployed in practice. Specifically, we have described that the lightweight design of LightFAD-DETR allows deployment on portable devices, mobile terminals, and edge computing units, enabling real-time pest detection even in remote forestry areas. We have also discussed potential applications such as integration with unmanned aerial vehicles for large-scale forest surveys and with ground-based IoT monitoring stations for continuous pest surveillance.
This revision can be found in the revised manuscript on page 22, lines 763–774, with the new text highlighted in red.
Comments 7:
In the discussion section, it is necessary to pay more attention to comparing the developed model with previously known ones. What is the scientific and practical value of the obtained model? This should be clearly stated.
Response 7:
Thank you for this valuable suggestion. We agree with the reviewer’s comment and have expanded the Discussion section to more explicitly compare the proposed LightFAD-DETR with previously known models. In particular, we emphasized that traditional approaches often sacrifice efficiency for accuracy or vice versa, whereas our model achieves a more balanced trade-off by integrating lightweight architecture with enhanced feature aggregation for small-object recognition.
Furthermore, we have clarified both the scientific and practical value of the model. From a scientific perspective, LightFAD-DETR demonstrates how re-parameterization and adaptive feature modeling can be combined to address challenges such as complex backgrounds and tiny pest targets in economic forests. From a practical perspective, the model offers a deployable and efficient solution for real-time forestry pest monitoring on resource-constrained devices.
This revision can be found in the revised manuscript on page 22, lines 748–762, with the updated text marked in red.
Comments 8:
It is advisable to indicate the main quantitative characteristics of the developed model in the conclusions.
Response 8:
Thank you for this helpful suggestion. We agree with the reviewer’s comment and have revised the Conclusion section to include the main quantitative characteristics of the proposed model. Specifically, we added the model’s detection accuracy (mAP), inference speed (FPS), and computational efficiency (parameters and FLOPs) to highlight its performance more explicitly.
This revision can be found in the revised manuscript on page 23, lines 822–827, with the new text highlighted in red.
Comments 9:
References to sources from which calculation formulas have been taken must be indicated in the text of the article. Alternatively, authors must indicate that these formulas are their own.
Response 9:
Thank you for pointing this out. We agree with the reviewer’s suggestion. Accordingly, we have revised the manuscript to provide explicit references for the formulas where they were taken from the literature. In the revised version, references have been added to Formula 1 through Formula 12 as appropriate.
This revision can be found in the revised manuscript on pages 8–11, formulas 1–12, where the corresponding sources are now cited in the text, with the added references marked in red.
Comments 10:
Figure 6a: Authors should add a description of the colour labels in this figure.
Response 10:
Thank you for this helpful suggestion. We agree with the reviewer’s comment and have revised the manuscript to add a clear description of the colour labels used in Figure 6a. Specifically, we clarified that the 31 different colours correspond to the pest categories labeled from 0 to 30 in the Forestry Pest Dataset.
This revision can be found in the revised manuscript on page 12, lines 470–473, with the new text highlighted in red.
Reviewer 2 Report
Comments and Suggestions for Authors
The paper Lightweight Pest Object Detection Model for Complex Economic Forest Tree Scenarios proposes a lightweight architecture integrated with feature aggregation diffusion, designed for complex economic forest scenarios.
The article merits publication in the current journal because this model achieves an excellent balance between detection accuracy and computational efficiency. But, I have a few comments to improve it, as follows:
Abstract: no comments.
Introduction: It is well written, but some sentences are missing citations, such as:
Lines 82-90.
The paper's main contributions have been described; however, the authors need to state the hypothesis clearly.
Some sentences are exhaustive, like Sections 2.1. Pest Object Detection (Lines 124-156); 2.2. Small Object Detection (Lines 158-195).
Thus, these paragraphs are too long.
I suggest that the authors transfer some information to the supplementary material. It's tiring to read the article in the current version.
The title of Table 1 is too short.
Table 2: Same comment as table 1.
Discussion: Although the discussion is short, I consider it enough.
Author Response
Comments 1:
Introduction: It is well written, but some sentences are missing citations, such as: Lines 82–90.
Response 1:
Thank you for this valuable comment. We agree with the reviewer’s suggestion. Accordingly, we have revised the Introduction to add the missing citations to support the statements in this part of the text.
This revision can be found in the revised manuscript on page 2, lines 87–91, where the newly added references are highlighted in red.
Comments 2:
The paper's main contributions have been described; however, the authors need to state the hypothesis clearly.
Response 2:
Thank you for this constructive suggestion. We agree with the reviewer’s comment and have revised the Introduction to clearly state the research hypothesis. Specifically, we added a statement to clarify that our hypothesis is that a lightweight detection model integrating re-parameterized multi-branch convolution, adaptive feature aggregation, and efficient normalization strategies can effectively improve the recognition of small pest objects in complex forestry scenes while maintaining real-time inference speed and reduced computational cost.
This revision can be found in the revised manuscript on page 3, lines 110–114, with the new text highlighted in red.
Comments 3:
Some sentences are exhaustive, like Sections 2.1. Pest Object Detection (Lines 124–156); 2.2. Small Object Detection (Lines 158–195). Thus, these paragraphs are too long. I suggest that the authors transfer some information to the supplementary material. It's tiring to read the article in the current version.
Response 3:
Thank you for this helpful suggestion. We agree that long paragraphs may affect readability. Accordingly, we have revised Section 2 by dividing it into shorter and clearer paragraphs to improve the flow and make it easier to read. However, we did not transfer the content into the supplementary material, because these descriptions provide essential context for understanding the novelty and contributions of the proposed method. We believe that keeping this information in the main text ensures completeness and accessibility for readers.
This revision can be found in the revised manuscript in Section 2, pages 4–5, where the paragraphs have been reorganized for improved clarity
Comments 4:
The title of Table 1 is too short.
Table 2: Same comment as table 1.
Response 4:
Thank you for this valuable suggestion. We agree with the reviewer’s comment and have revised the titles of both Table 1 and Table 2 to make them more detailed and self-explanatory. Specifically, Table 1 now clearly indicates that it presents the distribution of image samples across the 31 pest categories in the Forestry Pest Dataset, and Table 2 now specifies that it provides the experimental configuration details, including hardware, software, and training parameter settings.
These revisions can be found in the revised manuscript on page 12, line 469 (Table 1) and page 13, line 492 (Table 2), with the updated titles marked in red.
Reviewer 3 Report
Comments and Suggestions for Authors
The paper by Cheng et al is very interesting, I guess the main aim is to idenify forest pests with AI.
But as a non expert in this field of AI identification, I miss the link how the pictures are taken / generated and how the detections models will work. I can see that different aproaches were tested and the best option was found, but who is taking the images of the pupae in the forest an dhow is this related to a pest infestation? Traditional sampling is critisized with its limitations but with this new method I dont see any practical way, how this can work? Are the photos taken by drones? there are so many open questions and the journal is insects, not a journal for computer modeling or AI ananlysis, therefore the future way, how this can and will work needs to be explained.
I also see that there are no citations in the very short discussion but here you need to compare your results with the literature...
Almost all citations are from Asia/China, this is a bit strange. But maybe there is a bigger focus in this field there.
I am not sure how you use the term "lighweight" detection model and what is a "pest object"?
And what are complex ecosysthe forest tree scenarios? Most forest are very complex ecosysthems, but I dont think the paper investigated this.
In the species analysed it is not clear, what time of forst pest they are? wood borers?
no need to write related work in teh introduction as every introduction needs to present the important and related work of the past.
What pest is in the photo of fig 1? I cant recognise anything.
I think most readers of teh journal insects will need a lot more informations to understand this paper as today, most pests are still identified with traditional methods.
Author Response
Comments 1:
But as a non-expert in this field of AI identification, I miss the link how the pictures are taken / generated and how the detections models will work. I can see that different approaches were tested and the best option was found, but who is taking the images of the pupae in the forest and how is this related to a pest infestation? Traditional sampling is criticized with its limitations but with this new method I don’t see any practical way how this can work? Are the photos taken by drones? There are so many open questions and the journal is Insects, not a journal for computer modeling or AI analysis, therefore the future way, how this can and will work needs to be explained.
Response 1:
Thank you for these insightful comments. We agree that clarifying the dataset sources and the practical applicability of the model is essential, especially for readers with an entomological background. Accordingly, we have revised the manuscript in two ways:
In Section 4.1 (page 12, lines 451–467), we added details on how the images in the Forestry Pest Dataset were obtained, including their collection from field photographs, forestry pest monitoring websites, and expert-curated annotation processes. This ensures that the dataset reflects realistic forestry monitoring conditions and covers diverse pest developmental stages.
In the Discussion section (page 22, lines 763–774), we added a description of how the proposed model can be applied in real-life scenarios. Specifically, we clarified that due to its lightweight design, the model can be deployed on edge devices, mobile terminals, and unmanned aerial vehicles to conduct real-time monitoring in forests. This provides a practical alternative to traditional sampling, enabling large-scale, automated, and cost-effective pest surveillance.
Both revisions are marked in red in the manuscript.
Comments 2:
I also see that there are no citations in the very short discussion but here you need to compare your results with the literature...
Response 2:
Thank you for this valuable suggestion. We agree with the reviewer’s comment and have revised the Discussion section to provide explicit comparisons between our proposed model and previously reported works, supported by appropriate citations from the literature. These additions highlight both the scientific novelty and the practical advantages of LightFAD-DETR compared with known methods.
This revision can be found in the revised manuscript on page 22, lines 748–762, with the newly added comparisons highlighted in red.
Comments 3:
Almost all citations are from Asia/China, this is a bit strange. But maybe there is a bigger focus in this field there.
Response 3:
Thank you for this observation. We acknowledge that many of the references in our paper are from Asia, especially China. This reflects the fact that research on the application of artificial intelligence to forestry pest detection has been particularly active in China in recent years, with numerous datasets and benchmark models developed in this region. Therefore, the citation distribution corresponds to the current state of research in this field.
At the same time, we have ensured that relevant international studies have also been cited where applicable, so as to provide a comprehensive view of the field. We believe that this citation profile accurately represents the global research landscape of AI-based forest pest detection.
Comments 4:
I am not sure how you use the term "lightweight" detection model and what is a "pest object"?
Response 4:
Thank you for this valuable comment. We agree that these terms need to be clarified for readers who may not be familiar with computer vision terminology. Accordingly, we have revised the manuscript to provide explicit explanations. Specifically, on page 3, lines 94–96, we clarified that a “lightweight” detection model refers to a model with reduced parameters and computational complexity, making it more efficient and suitable for deployment on portable or resource-constrained devices. On page 4, lines 146–147, we defined a “pest object” as an instance of an insect pest in an image, which serves as the detection target for the model.
Both additions are marked in red in the revised manuscript.
Comments 5:
And what are complex ecosysthe forest tree scenarios? Most forest are very complex ecosystems, but I don’t think the paper investigated this.
Response 5:
Thank you for this helpful comment. We agree that the term needed clarification. Accordingly, we have revised the Introduction to explain what is meant by “complex economic forest ecosystems.” Specifically, we clarified that economic forests often consist of multiple tree species, layered canopy structures, and variable microclimatic conditions, which increase the difficulty of pest monitoring and detection.
This revision can be found in the revised manuscript on page 2, lines 60–64, with the new text highlighted in red.
Comments 6:
In the species analysed it is not clear, what type of forest pest they are? wood borers?
Response 6:
Thank you for this constructive comment. We agree that the types of pests represented in the dataset should be clarified. Accordingly, we have revised the manuscript to specify that the 31 pest species in the Forestry Pest Dataset cover several ecological groups, including wood borers, defoliators, sap-sucking pests, and leaf miners. This description makes it clearer to readers what kinds of pests are represented.
This revision can be found in the revised manuscript on page 12, lines 455–456, with the new text highlighted in red.
Comments 7:
No need to write related work in the introduction as every introduction needs to present the important and related work of the past.
Response 7:
Thank you for this comment. We would like to clarify that in our manuscript, the related work has been organized as a separate section (Section 2: Related Work), rather than being fully included in the Introduction. While the Introduction briefly mentions some important prior studies to set the context, Section 2 provides a more detailed discussion, especially focusing on Pest Object Detection and Small Object Detection, which are directly relevant to our research topic. We chose this structure to improve readability and to allow a more in-depth review of related methods without overloading the Introduction. This approach is also a common practice in academic writing, where related work is frequently presented as a standalone section.
This structure is retained in the revised manuscript, with Section 2 (pages 4–5) dedicated to related work, and the Introduction (pages 2–3) focusing primarily on background, motivation, and research objectives.
Comments 8:
What pest is in the photo of Fig. 1? I can’t recognise anything.
Response 8:
Thank you for this comment. We would like to clarify that Fig. 1 is primarily intended to illustrate the overall architecture of the proposed model and the process by which an input image is fed into the detection framework. The image included in Fig. 1 is an example drawn from the Forestry Pest Dataset, serving to demonstrate the data flow rather than to highlight recognition of a specific pest. For completeness, we note that the sample image depicts both male and female individuals of Drosicha contrahens, one of the pest species included in the dataset.
Round 2
Reviewer 1 Report
Comments and Suggestions for Authors
I have no comments.